# Characteristics of pulmonary multidrug-resistant tuberculosis patients in Tigray Region, Ethiopia: A cross-sectional study

Letemichael Negash Welekidan[1,2,3]*, Eystein Skjerve[2], Tsehaye Asmelash Dejene[3], Mengistu Welday Gebremichael[4], Ola Brynildsrud[1,5], Angelika Agdestein[6], Girum Tadesse Tessema[6], Tone Tønjum[7,8], Solomon Abebe Yimer[5,7]

1 Department of Para Clinical Sciences, Norwegian University of Life Sciences, Oslo, Norway, 2 Department of Production Animal Medicine, Norwegian University of Life Sciences, Oslo, Norway, 3 Department of Medical Microbiology and Immunology, Division of Biomedical Sciences, College of Health Sciences, Mekelle University, Mekelle, Ethiopia, 4 Department of Midwifery, College of Health Sciences, Mekelle University, Mekelle, Ethiopia, 5 Department of Bacteriology and Immunology, Norwegian Institute of Public Health, Oslo, Norway, 6 Section for Microbiology, Norwegian Veterinary Institute, Oslo, Norway, 7 Department of Microbiology, Unit for Genome Dynamics, University of Oslo, Oslo, Norway, 8 Department of Microbiology, Unit for Genome Dynamics, Oslo University Hospital, Oslo, Norway

* letemichael.negash@mu.edu.et

**Data Availability Statement:** All relevant data are within the manuscript and its Supporting Information files.

## Abstract

### Background

Tuberculosis (TB) is among the top 10 causes of mortality and the first killer among infectious diseases worldwide. One of the factors fuelling the TB epidemic is the global rise of multidrug resistant TB (MDR-TB). The aim of this study was to determine the magnitude and factors associated with MDR-TB in the Tigray Region, Ethiopia.

### Method

This study employed a facility-based cross-sectional study design, which was conducted between July 2018 and August 2019. The inclusion criteria for the study participants were GeneXpert-positive who were not under treatment for TB, PTB patients' ≥15 years of age and who provided written informed consent. A total of 300 participants were enrolled in the study, with a structured questionnaire used to collect data on clinical, sociodemographic and behavioral factors. Sputum samples were collected and processed for acid-fast bacilli staining, culture and drug susceptibility testing. Drug susceptibility testing was performed using a line probe assay. Logistic regression was used to analyze associations between outcome and predictor variables.

### Results

The overall proportion of MDR-TB was 16.7% (11.6% and 32.7% for new and previously treated patients, respectively). Of the total MDR-TB isolates, 5.3% were pre-XDR-TB. The proportion of MDR-TB/HIV co-infection was 21.1%. A previous history of TB treatment AOR 3.75; 95% CI (0.7–2.24), cigarette smoking AOR 6.09; CI (1.65–2.50) and patients who had

**Funding:** -LN is the award receiver. -CRPO/CHS/
PhD//MUNMBU/028/2010 -Royal Norwegian
Embassy, Addis Ababa, Ethiopia, an institutional
collaboration phase IV between the Norwegian
University of Life Sciences (Norway), Mekelle
University (Ethiopia) and Hawasa University
(Hawasa) -https://norad.no/en/toolspublications/
publications/2017/independent-mid-term-review-
of-institutional-collaboration-between-hawassa-
and-mekelle-universities-and-the-norwegian-
university-of-life-sciences/ -The funders had no
role in study design, data collection and analysis,
decision to publish, or preparation of the
manuscript.

**Competing interests:** The authors have declared
that no competing interests exist

**Abbreviations:** DR-TB, drug resistant tuberculosis;
FMOH, Federal Ministry of Health of Ethiopia; INH,
Isoniazid; LPA, Line probe assay; MDR-TB,
Multidrug-resistant tuberculosis; MTB,
*Mycobacterium tuberculosis*; Pre-XDR-TB, Pre-
extensively drug-resistant tuberculosis; PTB,
Pulmonary tuberculosis; RIF, Rifampicin; WHO,
World Health Organization.

an intermittent fever (AOR = 2.54, 95% CI = 1.21–5.4) were strongly associated with MDR-TB development.

## Conclusions

The magnitude of MDR-TB observed among new and previously treated patients is very alarming, which calls for an urgent need for intervention. The high proportion of MDR-TB among newly diagnosed cases indicates ongoing transmission, which suggests the need for enhanced TB control program performance to interrupt transmission. The increased proportion of MDR-TB among previously treated cases indicates a need for better patient management to prevent the evolution of drug resistance. Assessing the TB control program performance gaps and an optimal implementation of the WHO recommended priority actions for the management of drug-resistant TB, is imperative to help reduce the current high MDR-TB burden in the study region.

## Introduction

Tuberculosis (TB) is a chronic infectious disease, which is most commonly caused by *Mycobacterium tuberculosis (*MTB). TB has continued to be a major global public health concern, being one of the top 10 leading causes of death and the top killer among infectious diseases [1]. It is also the leading cause of death among people living with HIV/AIDS and the main cause of antimicrobial resistance-associated death [2]. Globally, there were an estimated 10 million new TB cases and 1.2 million deaths from TB in 2018 [3].

One of the major factors fuelling the TB epidemic is the emergence and spread of drug-resistant (DR) strains of MTB on new and previously treated cases, which creates a threatening and challenging condition for the prevention and control of TB [4]. Out of the total TB incident cases reported in 2018, 484,000 were resistant to rifampicin (RR-TB), and of these, 78% (3.4% new cases and 18% previously treated cases) had multidrug-resistant TB (MDR-TB). A total of 214,000 patients died due to MDR/RR-TB in 2018 [3].

TB is the most common cause of morbidity and mortality in Ethiopia. The country is among the three highest TB, TB/HIV and MDR-TB burden countries with estimates of 165,000 new TB cases and a rate of 151/100,000 population reported in 2018. In the same year, the number of cases of MDR/RR-TB was 1,600, the number of fatalities from TB was 24,000 for HIV-negative people and an additional 2,200 when including people living with HIV/AIDS [3].

Several studies in Ethiopia assessed the magnitude of drug-resistant TB. A review done by Biadglegne *et al.* reported that the occurrence of MDR-TB among TB patients in Ethiopia ranged from 3.3% to 46.3% [5]. Moreover, based on a recent meta-analysis report, the pooled estimate of MDR-TB among new and previously treated cases was 2% (1 to 2%) and 15% (12 to 17%), respectively [6]. Another study reported a MDR-TB prevalence ranging from 0 to 46.3% [7].

Previous studies in the different Regions of Ethiopia have shown variabilities in the magnitude of MDR-TB. The Oromia Region had 33.2% of MDR-TB cases [8], while in Jigjiga 10.2% of those smear-positive were MDR-TB patients [9]. In studies done in the Amhara Region, southwest Ethiopia and Addis Ababa, the magnitude of MDR-TB was reported to be 36.3%, 27.7% and 39.4%, respectively [10–12].

Early case detection and treatment of TB cases is essential to prevent and control drug-resistant TB. In the Tigray Region where this study was conducted, the treatment success rate was 80%, which is lower than the national and WHO target of achieving a 90% treatment success rate among the detected smear-positive cases. The treatment success rate observed in the Tigray Region was also lower than other regions such as the Afar Region in Ethiopia, which recorded a treatment success rate of 89% [13]. The major factor leading to a poor treatment success rate is treatment failure, which is mostly caused by drug-resistant (DR-TB) strains [13, 14].

In the Tigray Region, there is limited to no data on the burden and associated factors of DR-TB. Therefore, this study aimed at assessing the drug susceptibility pattern of TB for the first- and second-line anti-TB drugs and associated factors among TB patients in the Tigray Region, Ethiopia.

## Material and methods

### Study area

The study was conducted in six hospitals: the Alamata Hospital, Southern Zone; the Mekelle Hospital, Mekelle Special Zone; the Adigrat Hospital, Eastern Zone; the Adwa Hospital, Central Zone; the Shire/Suhul Hospital, Northwestern Zone and the Humera/Kahsay Abera Hospital, Western Zone) of the Tigray Regional State. The region has an estimated total population of 5.13 million [15]. The region is administratively divided into 7 zones (one especial zone, Mekelle), 52 districts and 814 *Kebeles* (lowest administrative unit). The health infrastructure of the region includes 40 hospitals, 223 health centers and 710 health posts serving the population of Tigray and neighboring regions. All the hospitals and health centers are equipped with TB diagnostic facilities (acid-fast bacteria (AFB) smear microscopy) and some hospitals with additional GeneXpert TB diagnostic facility. The Tigray Health Research Institute (THRI) is the only facility providing TB culture and drug susceptibility testing in the Region. The Stop TB (Directly Observed Treatment Short-course (DOTS)) Strategy is being implemented to control TB in the Region. The DOTS-Plus program has been introduced in all health facilities. Currently, there are nine treatment initiation centers and 62 treatment follow-up centers for MDR-TB in the Region. Fig 1 provides the geographical information about the hospitals and zones of the study area.

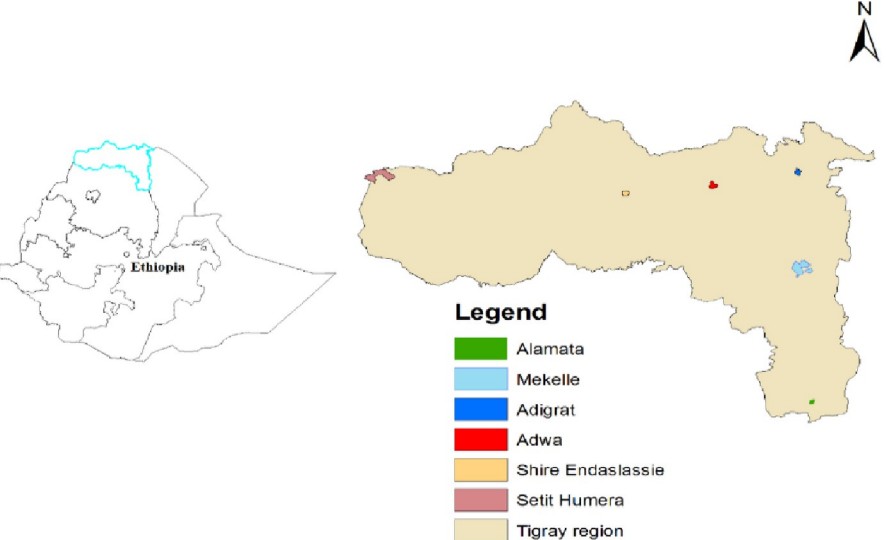

**Fig 1. Map of the study area in the Tigray Region, northern Ethiopia, July 2018 to August 2019.**

## Study design, population and inclusion criteria

The study was a hospital-based, cross-sectional study conducted from July 2018 to August 2019. The source population consisted of all presumptive pulmonary TB (PTB) cases in the Tigray Region during the study period. The study population were all presumptive PTB cases who visited the selected health facilities in the region during the study period. The inclusion criteria included PTB patients' who were not under treatment during the study, ≥15 years of age, who were GeneXpert-positive and could provide written informed consent, were included in the study. Critically ill patients from whom sociodemographic, clinical data and sputum samples could not be obtained, patients <15 years of age and extra-pulmonary TB cases were excluded from the study.

## Sample size determination

The sample size was determined by taking the required minimum number of MDR-TB patients to be enrolled in the study, which was estimated to be between 30–40 cases. This was to establish a sample with sufficient number of MDR-TB isolates to allow a sufficient power to find different varieties of MDR-TB isolates. Based upon an expected level of 10% [9] MDR-TB among TB patients, we aimed at recruiting at least 300 TB patients for the study. This would allow us to sample enough patients for the molecular studies, and also maintain a reasonable level of precision for the prevalence estimates. Without adjusting for clustering, the precision of the estimate would be at 6.7%-14.2% (95% CI, a relative precision of ± 3.3%).

Six government hospitals from the Tigray Region were selected based on the availability of the GeneXpert diagnostic technique as the primary test for presumptive PTB patients from the main zones of the Region without considering other factors (risk of high MDR-TB, high HIV and historically poor TB control program, etc.). The selected hospitals were the only hospitals with the GeneXpert test for the diagnosis of all presumptive PTB patients during the study. These hospitals were also giving diagnosis services for other neighboring health institutes that do not have a GeneXpert test. The standard practice of the hospitals was, first testing of all presumptive PTB patients by GeneXpert test, if the GeneXpert test result is susceptible TB the patient immediately started on 1st line-TB treatment and if the result is RIF-R TB the patient linked to MDR-TB clinic and immediately sputum sample used to be collected for culture before starting MDR-TB treatment.

However, for the purpose of this study, sputum samples were collected for culture and drug susceptibility testing from all consecutive GeneXpert positive PTB patients who fulfilled the inclusion criteria. A consecutive sampling technique was employed to recruit the study sample in all the hospitals until the required sample size was obtained.

## Operational definitions of variables

**New case:** A patient who has never had treatment for TB, or has been on treatment for less than four weeks.

**Previous treatment:** A patient who took TB treatment for one month or more in a previous time.

**Failure after treatment patient**: A patient who has previously been treated for TB, and whose treatment failed at the end of their most recent course of treatment.

**Defaulter (treatment after loss to follow-up):** A patient whose anti-TB treatment was interrupted for two consecutive months or more.

**Relapse:** A patient who has previously been treated for TB, was declared cured or whose treatment was completed at the end of their most recent course of treatment and is now diagnosed with a recurrent episode of TB.

**Mono-drug resistance**: Resistance to only one of the four first-line anti-TB drugs.

**Any drug resistance**: Resistance to one or more first-line anti-TB drugs.

**Multi-drug resistance**: Resistance to at least both isoniazid and rifampicin.

**Rifampicin resistance**: Resistance to rifampicin detected using phenotypic or genotypic methods, with or without resistance to other anti-TB drugs.

**Primary resistance (resistance among new TB cases)**: Resistance in patients who did not have a history of anti-TB treatment.

**Secondary resistance (resistance among previously treated cases)**: Resistance in patients previously treated with anti-TB drugs.

**Treatment success rate** refers to the percentage of notified TB patients who were successfully treated.

## Data collection

Socio-demographic and clinical data were collected from patients using a pre-tested structured questionnaire: Age, sex, residence, number of people living in a single room, history of imprisonment, history of TB treatment, contact history with TB patients in the past two years, and TB symptoms like cough, blood in sputum, fever, chest pain, shortness of breath, fatigue, night sweats, loss of appetite and body weight loss. Data on the history of previous medical illnesses and behavioral factors, including alcohol intake, cigarette smoking and Khat chewing habits, were also collected. HIV counselling and testing were performed for TB patients based on the recommendations of the Federal Ministry of Health of Ethiopia (FMOH) testing algorithm [16] at the study hospitals during the study period.

## GeneXpert® MTB/RIF assay

From each patient, a 4 ml sputum sample was collected "on-the-spot" and treated with a sample reagent (SR) containing NaOH and isopropanol according to the recommendation of FMOH [17]. The SR was added in a 2 to 1 ratio of the sputum sample, which was then homogenized and incubated for 15 minutes at room temperature following the manufacturer's instructions (Cepheid, Sunnyvale, CA, USA) [18]. The treated samples were transferred into the cartridge, and the cartridge was loaded into the GeneXpert instrument. Moreover, the Xpert® MTB/RIF purifies and concentrates MTB from the sputum samples, isolates genomic material from the captured mycobacteria by sonication, and subsequently amplifies the genomic DNA by PCR. The process identifies MTB DNA and rifampicin resistance, thus inducing mutations in the RNA polymerase beta (rpoB) gene in the MTB genome in a real-time format using fluorescent probes called molecular beacons.

## Sputum collection for culturing

A 5–10 ml sputum sample was collected from every GeneXpert MTB/RIF assay-positive participant by laboratory personnel, using a coded and sterile 50 ml falcon tube according to the recommendation of FMOH [17].

All sampled sputa for each participant were properly packed and kept at 4°C for transportation in an ice bag to the Tigray Health Research Institute (THRI), according to the international standards of the WHO recommendation for transport of a biological substance; category B, UN-3373. Specimens arrived within four-five days of collection, and were processed within seven days from the time of first collection.

## Culture and identification

**Decontamination and sputum processing.** Sputum samples were digested using freshly prepared N-acetyl-L-cysteine (NALC) and decontaminated by NaOH (1%). Phosphate buffer

(PH 6.8) were added to neutralize NaOH, and dilutes the homogenate to lessen the viscosity and specific gravity. The homogenate was centrifuged at 3000g at 4ºC for 15 minutes [19]. The direct microscopic examination of sputum, and from culture for acid-fast bacteria (AFB) using the standard Ziehl-Neelsen staining, was done at THRI.

**Sputum culture.** The decontaminated supernatant decanted sputum sample was cultured on a Lowenstein-Jensen (LJ) egg medium and on a liquid culture Mycobacterium Growth Indicator Tube; BACTEC MGIT 960 culture (Becton Dickinson Microbiology systems, Sparks, MD, USA), following the standard operational procedures. The tubes for the solid culture were incubated at 37˚C in a slant position to ensure an even distribution of inoculums for one week and thereafter at 37˚C in air for another seven weeks, and then checked once a week for mycobacterial growth. Cultures were considered negative when no colonies or growth were seen after eight weeks of incubation for a solid culture and six weeks (42 days) for a liquid culture. The growth of mycobacteria were confirmed by its typical colony morphology, acid-fast bacilli (AFB) staining, Capilia antigen test and inoculation onto a blood agar plate to rule out contamination.

## Drug susceptibility testing for first-line and second-line anti-TB drugs using Line Probe Assay (LPA)

Drug-susceptibility testing for first-line anti-TB drugs (isoniazid and rifampicin) using Geno-Type® MTBDRplus, and second line anti-TB drugs (ofloxacin, levofloxacin, moxifloxacin, amikacin, capromycin, kanamycin and viomycin) using GenoType MTBSL, was carried out by line probe assay genotypic method following the manufacturer's instruction (GenoType® MTBC; Hain Life Science, Nehren, Germany).

DNA extraction form culture was done using GenoLyse® kit (A and B) in a DNA contaminating free working area. After centrifugation, the supernatant was transferred to a new cryotube. In another room free from contaminating DNA, amplification Mixes A (10 μl) and B (35 μl) were prepared freshly and a total of 45 μl Master Mix was transferred to each polymerase chain reaction (PCR) tubes. The DNA extract (5 μl DNA) was added to respective PCR tubes, and 5 μl of DNA extract from H37Rv quality control strain to the positive control tube and 5 μl of distilled water to negative control tube was added. After amplification, the amplicon was detected with a series of procedures by adding different reagents to the strip. The strips formed color bands after the addition of the final substrate reagent [20].

## Quality assurance and quality control

All laboratory analyses were carried out following standard operating procedures. Both the solid culture and LPA procedures were checked and validated. Reference strains of MTB H37Rv were used as quality control organisms throughout the LPA test. Moreover, both the start and end controls were used during each batch of specimen processing and DNA extraction, as well as no template control being used for LPA reagents.

## Data entry and statistical analysis

Data were double-checked for completeness and cleaned before entry. A data missed by some respondents were traced back to the participants and completed the missed data. A few data which were not very important and missed by most respondents were omitted from the analysis as a whole. If important data were missed by some participants and we could not complete it by tracing back, we included only the respondents in the analysis. The complete and cleaned data were entered using the Epidata 3.1 data entry software. After cleaning and validation, the data were transferred into Stata (Stata SE 15/ SE for Windows, StataCorp, College Station, TX)

software for further statistical analysis. The study participants were categorized into two groups: patients with non-MDR-TB and patients with MDR-TB. Descriptive statistics were computed, and frequencies and proportions were presented in tables. Further statistical analyses were performed using a univariable and multivariable logistic regression to identify associations between sociodemographic, behavioral and clinical factors with the main outcome variable (MDR-TB). Results from the logistic regression analysis were presented using Odds Ratios (OR), with 95% Confidence Intervals and p-values. Variables with a $p < 0.20$ in the univariable analyses were included in the multivariable models. Models were built using a backward selection principle, with a likelihood-ratio (LR) test of $p = 0.10$ as a cut-of-point for excluding variables. A model fit was assessed using the Hosmer-Lemeshow test and a graphical examination using the sensitivity/specificity and receiver operating characteristic (ROC) curves.

### Ethical considerations

Ethical clearance was obtained from Mekelle University, College of Health Sciences Ethical Review and Research Committee (ERC 1438/2018), Ministry of Science and Higher Education, Ethiopia (SHE/S.M/14.4/708/19) and the Regional Committee for Medical Research Ethics in Eastern Norway (REK Øst) (2018/1118/REK sør-øst A). Written informed consent was secured from all study participants before the commencement of the study.

## Results

A total of 6322 presumptive PTB patients were excluded from the study because of the exclusion criteria stated. Primarily they were GeneXpert negative. In the present study, 300 GeneXpert® MTB/RIF assay positive study participants were included. Of these, 227 (75.67%), six (2%), 60 (20%) and seven (2.3%) were MTB culture-positive, non-*Tuberculosis mycobacterium* (NTM), culture-negative and contaminated, respectively. In the rest of this paper, only the 227 study participants with culture-positive tests were used for further analysis. The flow of the study participants' recruitment process is depicted in Fig 2.

The median age of all 300 participants was 30 years, ranging from 15–85 years. Of these, 196 (65.3%) were males and 104 (34.7%) females. A total of 220 (73.3%) participants were new TB cases, and 80 (26.7%) were previously treated patients. Furthermore, a majority of the participants 112 (37.3%) were in the age group from 25–34 years, 140 (46.7%) were married, 167 (55.7%) were urban dwellers, 98 (32.7%) had only elementary school, 84 (28.0%) were self-employed and 95 (31.7%) had no monthly income.

### Drug resistance patterns of first- and second-line anti-TB drugs

As presented in Table 1, a total of 40 (17.6%) of the 227 isolates were resistant to RIF by Geno-Type MTBDRplus assay while 42 (18.5%) were resistant to RIF by Xpert® MTB/RIF assay. Moreover, six (3.2%) RIF resistant isolates using Xpert® MTB/RIF assay were non MDR-TB by the Geno Type MTBDRplus method, and 2/38 (5.3%) RIF susceptible by Xpert® MTB/RIF assay were MDR-TB using the Geno Type MTBDRplus method.

Table 2 shows the pattern of resistance among 227 MTB isolates. The dominant isolates 189 (83.3%) were susceptible to INH and/or RIF, two (0.9%) had a mono resistance to RIF, three (1.3%) a mono resistance to INH, 43 (18.9%) a resistance to INH and/or RIF and 41 (18.1%) an overall resistance to INH. The proportion of MDR-TB among new and previously treated patients was 20 (11.6%) and 18 (32.7%), respectively, and the overall MDR-TB was 38 (16.7%). In this study, two (5.3%) of the MDR-TB isolates were pre-XDR-TB, which were resistant to fluoroquinolones (FQs), one of the second-line anti TB drugs. However, for the rest of second-line anti-TB drugs, the MDR-TB isolates were susceptible.

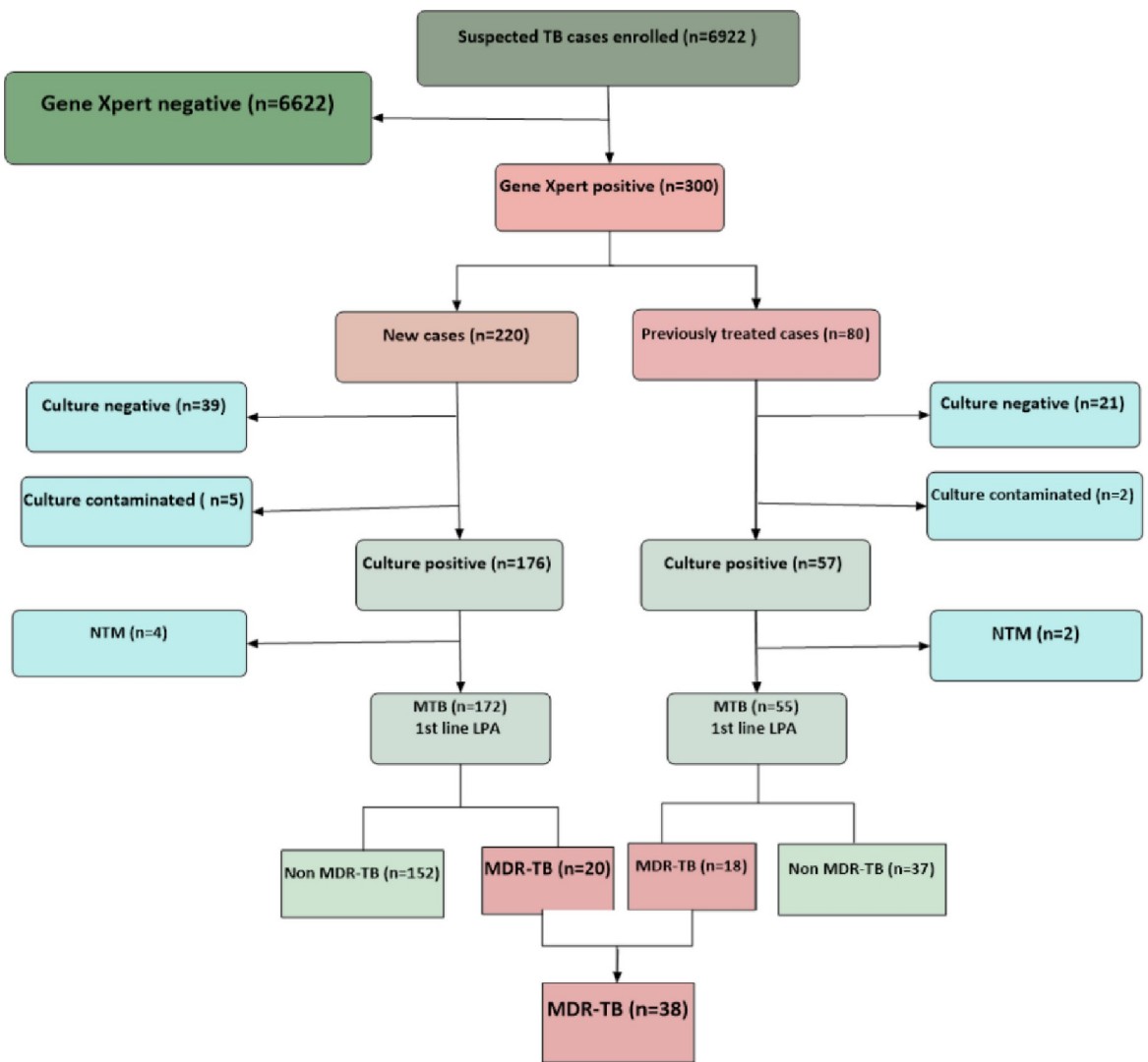

**Fig 2. Flow of the procedure followed for all patients recruited in the study in the Tigray Region, northern Ethiopia, July 2018 to August 2019.**

The proportion of overall TB/HIV, non-MDR-TB/HIV and MDR-TB/HIV co-infection among the 227 TB patients was 41 (33%), 33 (17.5%) and eight (21.1%), respectively. The Zonal distribution showed that MDR-TB was highest in the Southern Zone (21.1%), followed by the Mekelle Special Zone (18.4%) (Fig 3).

**Table 1. Rifampicin resistance patterns of isolates based on MTB status category of study participants in the Tigray Region, northern Ethiopia, July 2018 to August 2019.**

| | | Rifampicin resistance pattern | | |
| --- | --- | --- | --- | --- |
| | | Genotype MTBDR plus | | |
| | | Non MDR-TB (n = 189) | | MDR-TB (n = 38) |
| | | Susceptible, F (%) | Resistant (%) | Resistant (%) |
| GeneXpert | Susceptible | 183 (96.8) | 0 | 2 (5.3) |
| | Resistant | 4 (2.1) | 2 (1.1) | 36 (94.7) |

**Table 2. Drug resistance patterns of isolates to first- and second-line anti-TB drugs with TB patient category of study participants in the Tigray Region, northern Ethiopia, July 2018 to August 2019.**

| First-line resistance pattern | New cases (n = 172), F (%) | | Previously treated cases (n = 55), F (%) | Total cases (n = 227), F (%) |
|---|---|---|---|---|
| Any-S | 152 (88.4) | | 37 (67.3) | 189 (83.3) |
| RIF-R | 21 (12.2) | | 19 (34.5) | 40 (17.6) |
| RIF mono- R | 1 (0.6) | | 1 (1.8) | 2 (0.9) |
| RIF-S | 151 (87.8) | | 36 (65.5) | 187 (82.4) |
| INH-R | 23 (13.4) | | 18 (32.7) | 41 (18.1) |
| INH mono-R | 3 (1.7) | | 0 (0) | 3 (1.3) |
| INH-S | 149 (86.6) | | 37 (67.3) | 186 (81.9) |
| Any-R | 24 (14) | | 19 (34.5) | 43 (18.9) |
| MDR | 20 (11.6) | | 18 (32.7) | 38 (16.7) |
| **Second-line resistance pattern (38)** | **New cases (20), F (%)** | | **Previously treated cases (18), F (%)** | **Total cases (38), F (%)** |
| FLQ | R | 0 | 2 (11.1) | 2 (5.3) |
| | S | 20 (100) | 16 (88.9) | 36 (94.7) |
| AMK, CAP, KAN | R | 0 | 0 | 0 |
| | S | 20 (100) | 18 (100) | 38 (100) |
| KAN, CAP, VIO | R | 0 | 0 | 0 |
| | S | 20 (100) | 18 (100) | 38 (100) |
| KAN, AMK, CAP, VIO | R | 0 | 0 | 0 |
| | S | 20 (100) | 18 (100) | 38 (100) |
| Low-level KAN | R | 0 | 0 | 0 |
| | S | 20 (100) | 18 (100) | 38 (100) |

R = resistant, S = susceptible, FLQ = Fluoroquinolones (Ofloxacin, levofloxacin, Moxifloxacin), AMK = Amikacin, KAN = kanamycin, CAP = Capromycin, VIO = Viomycin.

### Factors associated with MDR-TB

Patients' sociodemographic parameters, risk behaviors, MDR-TB/HIV co-infection status, contact history and clinical presentations were compared between non-MDR and MDR-TB cases using univariable and multi-variable logistic regression analysis (Tables 3–5).

A history of previous TB treatment showed a strong association (AOR = 4.26, 95% CI = 1.99–9.14) with MDR-TB development, and patients with cigarette smoking habits (AOR = 6.09, 95% CI = 1.65–22.50) were more likely to develop MDR-TB compared to those who did not smoke. Patients who had an intermittent fever (AOR = 2.54, 95% CI = 1.21–5.4) were two times more likely to develop MDR-TB compared to those who did not experience a fever. Surprisingly, patients who used to consume alcohol (AOR = 0.28, 95% CI = 0.10–0.77) were less likely to develop MDR-TB compared to those who did not take alcohol.

No sociodemographic characteristics and co-infections status were associated with MDR-TB development. This included an absence of association between MDR-TB and HIV infection. In the univariable analysis, patients with a duration of symptoms > 60 days were more likely to develop MDR-TB, but this variable did not show a significant association in the multivariable model.

### Discussion

This is the first and largest study attempting to assess the magnitude and associated factors of MDR-TB in the Tigray Region of Ethiopia. A periodic assessment of the prevalence and associated factors of drug resistance in high TB burden countries is essential to identify early- and

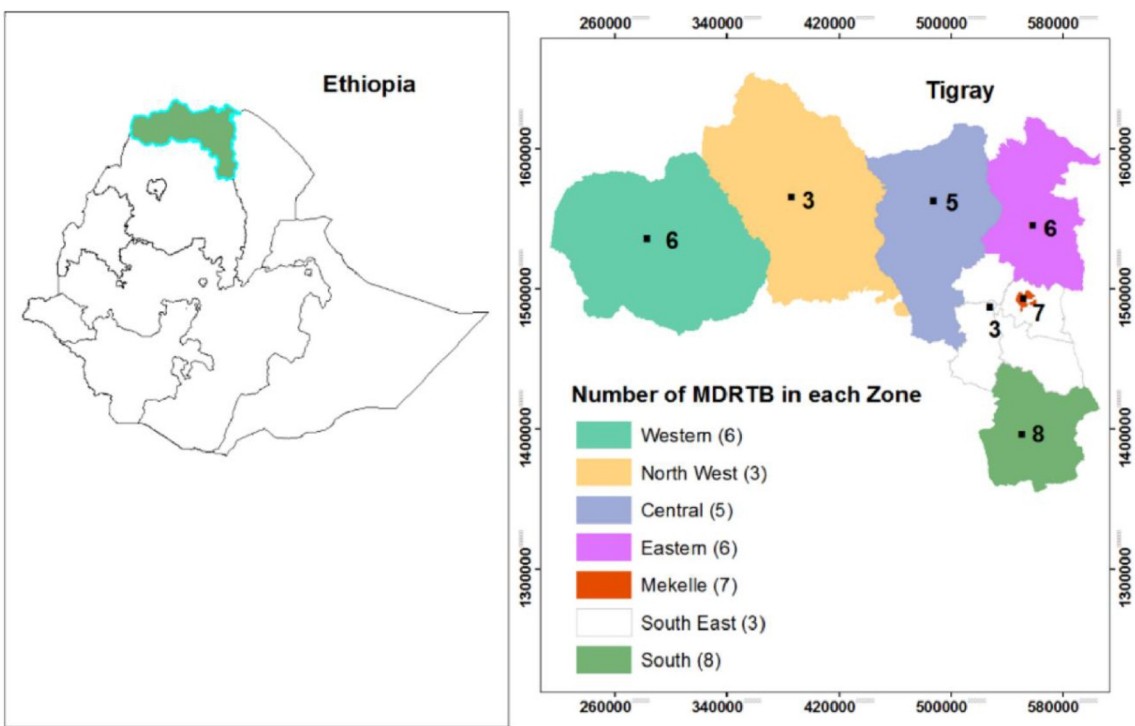

**Fig 3. Zonal distribution of 38 MDR-TB among TB patients (N = 227) in the Tigray Region, northern Ethiopia, July 2018 to August 2019.**

address the challenges of drug-resistant TB transmission. This helps to enhance the TB control program performance and achieve the End TB Strategy goals.

The overall proportion of MDR-TB observed in our study was 16.2% (11.6% among new cases and 32.7% among previously treated cases). The WHO data indicates that 6–10% of new TB cases and 13–60% of cases in previously treated patients are MDR [21]. The overall proportion of MDR-TB observed in our study was higher compared to previous studies conducted in Addis Ababa, Ethiopia [22, 23], northwest Ethiopia [24, 25], Jigjiga in the Somali Region [9] and the Oromia Region [26], which showed an overall prevalence of MDR-TB of 11.5%, 5.6%, 10.2%, 1.8%, 1.8% and 4.7%, respectively. The finding was also higher than the study done in China [27], Kenya [28], Tanzania [29], India [30], Vietnam [31], Lima Peru [32], Dalian China [33] and northeastern China [34], where the overall MDR-TB prevalence was estimated at 11.3%, 4.8%, 6.3%, 5.6%, 6.9%, 6.6%, 10.1%, 8.7%, respectively. Conversely, the overall MDR-TB proportion observed in our study was lower than former study findings, which showed a 39.4%, 27.2%, 39.2% and 26% overall MDR-TB prevalence reported from Addis Ababa, Ethiopia [12], southwest Ethiopia [10], Uganda [35] and Taiwan [36], respectively. The increased proportion of MDR-TB observed in our study is very alarming to the TB/MDR-TB control program in the study area, as well as for the country at large.

Increased MDR-TB among new TB cases is an indicator of the ongoing transmission of drug-resistant TB. The proportion of new MDR-TB (11.6%) cases observed in our study is higher than many other studies conducted in Ethiopia, such as eastern Ethiopia (1.1%) [37], northwest Ethiopia (2.3%) [24], Jigjiga, in the Somali Region (4.5%) [9] and the Amhara Region (1.0%) [38]. The finding is also higher than studies conducted in other countries: Tanzania (4.3%) [29], India (2.9%) [30], Vietnam (4.2%) [31], Dalian China (5.8%) [33] and

**Table 3.** Sociodemographic and behavior factors associated with MDR-TB among the study participants in the Tigray region, northern Ethiopia, July 2018 to August 2019.

| Variables | Non-MDR (189), F (%) | MDR (38), F (%) | COR (95% Cl) | P-value | AOR (95% Cl) | P-value |
|---|---|---|---|---|---|---|
| **Sex** | | | | | | |
| Male | 122 (64.6%) | 22 (57.9) | 1 | | - | - |
| Female | 67 (35.5) | 16 (42.1) | 1.32 (0.65–2.69) | 0.438 | | |
| **Age in years** | | | | | | |
| 15–24 | 38 (20.1) | 9 (23.7) | 1 | | | |
| 25–34 | 76 (40.2) | 14 (36.8) | 0.78 (0.3–1.96) | 0.594 | - | - |
| 35–44 | 36 (19.1) | 10 (26.3) | 1.17 (0.43–3.22) | 0.757 | | |
| 45–54 | 20 (10.6) | 2 (5.3) | 0.42 (0.08–2.14) | 0.298 | | |
| ≥55 | 19 (10.1) | 3 (7.9) | 0.67 (0.16–2.75) | 0.575 | | |
| **Residence** | | | | | | |
| Urban | 103 (54.5) | 24 (63.2) | 1 | | | |
| Rural | 86 (45.5) | 14 (36.8) | 0.7 (0.34–1.43) | 0.328 | - | - |
| **Marital status** | | | | | | |
| Single | 90 (47.6) | 15 (39.5) | 1 | | | |
| Married | 86 (45.5) | 17 (44.7) | 1.18 (0.56–2.52) | 0.658 | | |
| Divorced | 4 (2.1) | 2 (5.3) | 3 (0.50–17.85) | 0.227 | - | - |
| Widowed | 9 (4.8) | 4 (10.5) | 2.67 (0.73–9.77) | 0.139 | | |
| **Pregnant (48)** | | | | | | |
| No | 8 (80) | 37 (97.4) | 1 | | | |
| Yes | 2 (20) | 1 (2.6) | 2.53 (0.22–28.6) | 0.454 | - | - |
| **Lactating (48)** | | | | | | |
| No | 9 (90) | 37 (97.4) | 1 | | | |
| Yes | 1 (10) | 1 (2.6) | 5.08 (0.31–3.07) | 0.254 | - | - |
| **Family size** | | | | | | |
| ≤3 | 109 (57.7) | 19 (50) | 1 | | | |
| ≥4 | 80 (42.3) | 19 (50) | 1.36 (0.68–2.74) | 0.385 | - | - |
| **Education** | | | | | | |
| Illiterate | 60 (31.8) | 10 (26.3) | 1 | | | |
| 1–8 grades | 58 (30.7) | 16 (42.1) | 1.66 (0.69–3.95) | 0.256 | - | - |
| 9–12 grades | 54 (28.6) | 8 (21.1) | 0.89 (0.33–2.42) | 0.817 | | |
| Diploma and above | 17 (9) | 4 (10.5) | 1.41 (0.39–5.07) | 0.597 | | |
| **Occupation** | | | | | | |
| Housewife | 20 (10.6) | 5 (13.2) | 1 | | | |
| Farmer | 46 (24.3) | 10 (26.3) | 0.87 (0.26 2.87) | 0.819 | - | - |
| Self-employed | 58 (30.7) | 8 (21.1) | 0.55 (0.16–1.88) | 0.342 | | |
| Government employee | 10 (5.3) | 2 (5.3) | 0.8 (0.13–4.87) | 0.809 | | |
| Student | 13 (6.9) | 4 (10.5) | 1.23 (0.28–5.45) | 0.785 | | |
| No work | 42 (22.2) | 9 (23.7) | 0.86 (0.25–2.89) | 0.804 | | |
| **Monthly income (Birr)** | | | | | | |
| <500 | 41 (21.7) | 11 (29) | 1 | | | |
| 500–2000 | 48 (25.4) | 10 (26.3) | 0.78 (0.3–2.01) | 0.603 | | |
| >2000 | 39 (20.6) | 4 (10.5) | 0.38 (0.11–1.30) | 0.124 | - | - |
| No means of income | 61 (32.3) | 13 (34.2) | 0.79 (0.32–1.94) | 0.614 | | |
| **Smoking** | | | | | | |
| No | 176 (93.1) | 32 (84.2) | 1 | | 1 | |
| Yes | 13 (6.9) | 6 (15.8) | 2.54 (0.9–7.17) | 0.079 | 6.09 (1.7–22.5) | **0.007** |

(*Continued*)

**Table 3.** (Continued)

| Variables | Non-MDR (189), F (%) | MDR (38), F (%) | COR (95% Cl) | P-value | AOR (95% Cl) | P-value |
|---|---|---|---|---|---|---|
| **Alcohol intake** | | | | | | |
| No | 111 (58.7) | 28 (73.7) | 1 | | 1 | |
| Yes | 78 (41.3) | 10 (26.3) | 0.51 (0.23–1.11) | 0.088 | 0.28 (0.1–0.77) | **0.014** |
| **Khat chewing** | | | | | | |
| No | 173 (91.5) | 33 (86.8) | 1 | | | |
| Yes | 16 (8.5) | 5 (13.2) | 1.64 (0.56–4.78) | 0.366 | - | - |

northeastern China (4.2%) [34]. The high proportion of MDR-TB among newly diagnosed TB patients observed in this study is of high concern, which requires an urgent intervention to improve the quality of TB control to interrupt the transmission of drug-resistant TB [39].

The proportion of MDR-TB (32.7%) among previously treated cases in our study was more or less similar with a study report from southwest Ethiopia [40], which showed an MDR prevalence of 31.4%. However, our report was lower than that reported from the St. Peter's TB Specialized Hospital, Addis Ababa, Ethiopia, which showed an MDR-TB prevalence of 58% [41]. This may not be surprising as the St-Peter's Specialized Hospital in Addis Ababa is the national referral hospital for specialized TB care in the country. Conversely, our finding was higher than several other previous study results reported from Jigjiga, the Somali Region, Ethiopia [9], Dalian China [33], northeastern China [34] and Vietnam [31], which showed an MDR prevalence of 22.6%, 17.7%, 27.6% and 23.1%, respectively. The increased MDR-TB proportion among previously treated cases in our study indicates a need for better patient management to help prevent the evolution of resistance in the study area.

Five percent of the cases in our study had pre-XDR-TB or FQs-resistant MDR-TB, which is higher than a recent study conducted in Ethiopia (3.4%) [42] and China (0.7%) [34]. Studies in

**Table 4. Associations of TB/HIV co-infection and contact history with MDR-TB among study participants in the Tigray region, northern Ethiopia, July 2018 to August 2019.**

| Variables | Non-MDR-TB (%) (n = 189) | MDR-TB (%) (n = 38) | OR (95% Cl) | P-value |
|---|---|---|---|---|
| **TB patient history in family** | | | | |
| No | 144 (76.2) | 26 (68.4) | 1 | |
| Yes | 45 (23.8) | 12 (31.6) | 1.48 (0.69–3.16) | 0.316 |
| **History of TB close contact** | | | | |
| No | 149 (78.8) | 25 (65.8) | 1 | |
| Yes | 40 (21.2) | 13 (34.2) | 1.94 (0.91–4.12) | 0.086 |
| **History of MDR-TB contact** | | | | |
| No | 176 (93.1) | 33 (86.8) | 1 | |
| Yes | 13 (6.9) | 5 (13.2) | 2.05 (0.69–6.14) | 0.199 |
| **Diabetes mellitus status** | | | | |
| No | 183 (96.8) | 36 (94.7) | 1 | |
| Yes | 6 (3.2) | 2 (5.3) | 1.69 (0.33–8.73) | 0.528 |
| **HIV status** | | | | |
| Negative | 156 (82.5) | 30 (79) | 1 | |
| Positive | 33 (17.5) | 8 (21) | 1.26 (0.53–3) | 0.600 |
| **History of prison** | | | | |
| No | 163 (86.2) | 34 (89.5) | 1 | |
| Yes | 26 (13.8) | 4 (10.5) | 0.74 (0.24–2.25) | 0.593 |

**Table 5. Associations of clinical presentations with MDR-TB among study participants in the Tigray Region, northern Ethiopia, July 2018 to August 2019.**

| Variables | Non-MDR-TB (189), F (%) | MDR-TB (%) (38), F (%) | OR (95% Cl) | P-value | AOR (95% Cl) | P-value |
|---|---|---|---|---|---|---|
| **TB history** | | | | | | |
| No | 152 (80.4) | 20 (52.6) | 1 | | 1 | |
| Yes | 37 (19.6) | 18 (47.4) | 3.7 (1.78–7.68) | **0.001** | 4.26 (1.99–9.14) | **<0.001** |
| **Duration of illness in days (198)** | | | | | | |
| ≤60 | 104 (62.6) | 10 (31.3) | 1 | | | |
| >60 | 62 (37.4) | 22 (68.7) | 3.69 (1.64–8.30) | **0.002** | - | - |
| **Body mass index** | | | | | | |
| >18.5 | 53 (28) | 8 (21) | 1 | | | |
| ≤18.5 | 136 (72) | 30 (79) | 0.68 (0.29–1.59) | 0.377 | - | - |
| **Weight loss** | | | | | | |
| No | 18 (9.5) | 6 (15.8) | 1 | | | - |
| Yes | 171 (90.5) | 32 (84.2) | 0.56 (0.21–1.52) | 0.257 | - | |
| **Chest pain** | | | | | | |
| No | 57 (30.2) | 7 (18.4) | 1 | | | - |
| Yes | 132 (69.8) | 31 (81.6) | 1.91 (0.8–4.6) | 0.147 | - | |
| **Coughing for > 2 weeks** | | | | | | |
| No | 27 (14.3) | 6 (15.8) | 1 | | | |
| Yes | 162 (85.7) | 32 (84.2) | 0.89 (0.34–2.33) | 0.810 | - | - |
| **Shortness of breath** | | | | | | |
| No | 92 (48.7) | 19 (50) | 1 | | | |
| Yes | 97 (51.3) | 19 (50) | 0.95 (0.47–1.90) | 0.882 | - | - |
| **Hemoptysis** | | | | | | |
| No | 172 (91) | 33 (86.8) | 1 | | | |
| Yes | 17 (9) | 5 (13.2) | 1.53 (0.53–4.44) | 0.431 | - | - |
| **Intermittent fever** | | | | | | |
| No | 119 (63) | 17 (44.7) | 1 | | 1 | |
| Yes | 70 (37) | 21 (55.3) | 2.1 (1.04–4.25) | **0.039** | 2.54 (1.21–5.4) | **0.014** |
| **Night sweats** | | | | | | |
| No | 44 (23.3) | 9 (23.7) | 1 | | | |
| Yes | 145 (76.7) | 29 (76.3) | 0.98 (0.43–2.22) | 0.957 | - | - |
| **Appetite loss** | | | | | | |
| No | 68 (36) | 9 (23.7) | 1 | | | |
| Yes | 121 (64) | 29 (76.3) | 1.81 (0.81–4.05) | 0.148 | - | - |
| **Fatigue and malaise** | | | | | | |
| No | 142 (75.1) | 30 (79) | 1 | | | |
| Yes | 47 (24.9) | 8 (21) | 0.81 (0.35–1.88) | 0.617 | - | - |

other countries showed a very high proportion of pre XDR-TB compared to our study result: Morocco (22.2%) [43], China (14.5%) [27], Vietnam (17.9%) [31], India (27.6%) [44], western India (22%) [45] and Pakistan (38.7%) [46]. Some of the factors associated with the high pre-XDR prevalence in those countries were related to a widespread use of FQs without prescriptions for the treatment of new TB cases and other undiagnosed respiratory infections in the private sectors [31, 45], with most of the study participants being treatment failures and chronic cases [45]. Overall, the proportion of pre-XDR-TB observed in our study is of great concern, thereby indicating the possibility of an increasing resistance to second-line anti-TB drugs in the region.

Previous studies indicated that several factors such as a previous history of TB treatment (failures, defaulters and relapses), contact with MDR-TB patients and water pipe smoking [23, 46, 47] are all associated with the development of MDR-TB. In our study, patients with a previous history of TB treatment were more likely to develop MDR-TB, with similar findings reported from other studies in Ethiopia and several other countries [6, 36, 40, 48–52]. This association might be related to unsatisfactory/noncompliance by patients or clinicians to anti-TB treatment, a lack of supervision of treatment or poor quality of the DOTS program, improper drug regimens and an inadequate or irregular drug supply, which may potentiate genetic mutations in the bacteria, and can result in acquired drug resistance [48, 53–55].

Cigarette smoking was associated with MDR-TB. This is similar to other study findings [23, 25, 35]. Cigarette smoking directly causes ciliary dysfunction; this diminishes the immunity of individuals, which makes them prone to primary MDR-TB [56, 57].

A univariable analysis showed that patients who experienced TB symptoms for a duration of more than 60 days were more likely to develop MDR-TB compared to their counterparts. This finding is in line with studies and reports from several other countries [40, 48, 58]. One of the mechanisms of MTB drug resistance arises from spontaneous point mutations. Over time, these mutations accumulate, and acquired drug resistance may occur if diagnosis and treatment is not initiated early [59]. A delay in treatment can also result in an overgrowth of MTB (a high grade of sputum positivity), which causes a delayed sputum conversion during treatment, and in turn is associated with an acquired MDR-TB [58, 60, 61].

The current proportion of MDR-TB/HIV co-infection was 21.1%, which is lower than the findings in previous studies in northwest Ethiopia (28.6%) [24], Addis Ababa, Ethiopia (79.8%) [12] and southwestern Ethiopia (43.5%) [40]. Nevertheless, this is higher than the results reported from Mali (10.5%) [47], Thailand (4.8%) [58] and India (5.6%) [30]. HIV infection is the strongest risk factor for the development of active TB regardless of the type of MTB, whether drug-susceptible or primary drug-resistant, as it suppresses the immune system of the individuals [62]. In our study, HIV co-infection was not associated with the development of MDR-TB, which is similar to other findings [40, 47, 58], though some studies reported its association with MDR-TB [37, 38, 49]. A review by Wells *et al*. reported that there was no association between MDR-TB and HIV in several studies in Africa, Russia, Vietnam, India and other multi-country studies. However, this review reported that there was an association in a study done in the United States and in Ethiopia, but in this review the association was only with primary MDR-TB [62]. This review in particular reported that a specific genotype family of drug-resistant strains of MTB might play a role in transmission, especially among people living with HIV infection. The Beijing genotype family, which includes the "W" strain of MTB implicated in many MDR-TB outbreaks in the United States, is more virulent and is associated with an anti-TB drug resistance in specific geographic settings [62].

We were surprised to observe the association between the consumption of alcohol and a lower chance of acquiring MDR-TB. As far as our literature review is concerned, there is no study that supports our finding. Our hypothesis is that this may have happened by chance. Further study is warranted to understand the link between alcohol intake and drug resistance.

In the current study, none of the sociodemographic factors were statistically associated with MDR-TB development, which is similar with other study reports [24, 58, 63]. A recent similar study in the Tigray Region reported that age was marginally associated with MDR-TB [64]. Yet, this result could not be reproduced in our study. Sociodemographic variables such as educational status [65], age [49] and residence [26] were associated with MDR-TB in studies conducted in the various other regions of Ethiopia.

Ethiopia has initiated an innovative community health program (health extension program) in all its regional states, including Tigray, over the last 15 years. This initiative can play a

big role in increasing early case detection and an improved treatment success rate by improving service access to TB patients [66, 67]. Regardless of the priceless role of health extension workers in TB/MDR-TB prevention and the control program, a study conducted in the Tigray Regional State reported that only one-fourth (25%) of the health posts were working efficiently [68]. This may contribute to a poor TB control program performance at the community level, which leads to a poor case detection and treatment outcome, and may ultimately contribute to an increased number of treatment failure cases and the emergence and transmission of drug-resistant TB. A previous study showed that a high proportion of defaulters was reported in the Tigray Region compared to other regions of Ethiopia [13]. In addition, another study in Tigray reported a low level of knowledge about the cause of TB and the consequences of a poor treatment adherence to its treatment [69]. Besides, a recent assessment report in Tigray Region indicates that only 21.7% of household contacts were screened for TB by health extension workers [70], which are by far lower than the target stated by the national TB program to screen all household contacts [71] and WHO target of greater than 90% household contacts to be screened for TB [3].

Several factors have contributed to the development and transmission of drug-resistant TB. Genetic diversities of drug-resistant isolates might be attributable to some host and environmental factors besides strain evolution in different geographic regions [72–75]. Drug resistance in MTB arises at a low frequency of spontaneous chromosomal mutations and inconsistent drug supply, suboptimal prescription and poor patient adherence [73, 74]. In particular, patients who have a previous TB treatment history such as treatment failures, defaulters or relapse cases are at a greater risk of developing MDR-TB [5, 75, 76]. The WHO underscores the importance of the proper management of MDR-TB to help address its global threat to human beings. To help alleviate this threat, the WHO has recommended five priority actions, including the prevention of the development of MDR-TB through a high-quality treatment of drug-susceptible TB, expanding the rapid testing and detection of drug-resistant TB, thereby providing immediate access to effective treatment and proper care, the prevention of transmission through infection control and an increase in political commitment and financing. High TB-burden countries like Ethiopia should be committed to optimally implementing these actions to reduce the emergence and transmission of drug-resistant TB in Tigray and all other regions of the country.

## Limitations of the study

One of the limitations of this study was that non-MDR-TB isolates were not tested for second-line anti-TB drugs, as other studies found that such types of isolates were highly resistant to second-line anti-TB drugs [44]. The other limitation was that extra-pulmonary patients were not included in the study. Hence, the authors recommend further study on the overall study of the drug susceptibility pattern of first- and second line anti-TB drugs for non-MDR and MDR-TB from pulmonary and extra-pulmonary patients, in order to obtain full information on the magnitude of DR-TB in the study area.

## Conclusion

The magnitude of MDR-TB observed among new and previously treated patients is very alarming, which calls for an urgent need for intervention. The high proportion of MDR-TB among newly diagnosed cases indicates an ongoing transmission, which suggests the need for an enhanced TB control program performance to interrupt transmission. The increased proportion of MDR-TB among previously treated cases indicates a need for better patient management to help prevent the evolution of drug resistance. The associated factors of this study indicate the need for consideration of these predisposing factors in the prevention and

intervention program of the Region. Overall, the findings highlight the importance of strengthening the Regional TB Control Program to detect and provide early appropriate treatment and follow-up for TB cases. Assessing the TB control program performance gaps in the region, and an optimal implementation of the five WHO-recommended priority actions for the management of drug-resistant TB, is imperative to reduce the current high MDR-TB burden in the study region.

## Supporting information

**S1 Data.**
(XLSX)

**S1 File. Questionnaire.**
(PDF)

## Acknowledgments

The authors are thankful to the Norwegian Veterinary Institute, the Tigray Health Research Institute, the Ethiopian Public Health Institute and the Armauer Hansen Research Institute for providing material and reagent support. We would also like to thank the study participants for their willingness to participate in the study. We are very grateful to all the data collectors at the selected hospitals for their help. The Tigray Regional Health Bureau and the administrators of each hospital are also greatly acknowledged for their cooperation.

## Author Contributions

**Conceptualization:** Letemichael Negash Welekidan, Eystein Skjerve, Tsehaye Asmelash Dejene, Solomon Abebe Yimer.

**Data curation:** Letemichael Negash Welekidan, Eystein Skjerve, Tsehaye Asmelash Dejene, Mengistu Welday Gebremichael, Solomon Abebe Yimer.

**Formal analysis:** Letemichael Negash Welekidan, Eystein Skjerve.

**Funding acquisition:** Letemichael Negash Welekidan.

**Investigation:** Letemichael Negash Welekidan.

**Methodology:** Letemichael Negash Welekidan, Eystein Skjerve, Tsehaye Asmelash Dejene, Solomon Abebe Yimer.

**Project administration:** Eystein Skjerve.

**Resources:** Eystein Skjerve.

**Software:** Letemichael Negash Welekidan, Eystein Skjerve.

**Supervision:** Eystein Skjerve, Tsehaye Asmelash Dejene, Solomon Abebe Yimer.

**Validation:** Letemichael Negash Welekidan, Eystein Skjerve.

**Visualization:** Letemichael Negash Welekidan.

**Writing – original draft:** Letemichael Negash Welekidan.

**Writing – review & editing:** Letemichael Negash Welekidan, Eystein Skjerve, Tsehaye Asmelash Dejene, Mengistu Welday Gebremichael, Ola Brynildsrud, Angelika Agdestein, Girum Tadesse Tessema, Tone Tønjum, Solomon Abebe Yimer.

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
