## [Decision Letter · Decision Letter 0]

22 Apr 2020

PONE-D-20-06975

Multidrug-resistant Tuberculosis and Its Associated Factors from Tuberculosis Patients in Tigray Region, Ethiopia: a cross sectional study

PLOS ONE

Dear Dr. Welekidan,

Thank you for submitting your manuscript to PLOS ONE. After careful consideration, we feel that it has merit but does not fully meet PLOS ONE’s publication criteria as it currently stands. Therefore, we invite you to submit a revised version of the manuscript that addresses the points raised during the review process.

We would appreciate receiving your revised manuscript. To enhance the reproducibility of your results, we recommend that if applicable you deposit your laboratory protocols in protocols.io, where a protocol can be assigned its own identifier (DOI) such that it can be cited independently in the future. For instructions see: http://journals.plos.org/plosone/s/submission-guidelines#loc-laboratory-protocols

We look forward to receiving your revised manuscript.

Kind regards,

Frederick Quinn

Academic Editor

PLOS ONE

Journal Requirements:

2. We noticed you have some minor occurrence(s) of overlapping text with the following previous publication(s), which needs to be addressed:

https://doi.org/10.1186/s12879-017-2389-6

https://doi.org/10.1155/2019/2923549

http://www.stoptb.org/wg/gli/assets/documents/gli_mycobacteriology_lab_manual_web.pdf

In your revision ensure you cite all your sources (including your own works), and quote or rephrase any duplicated text outside the Methods section. Further consideration is dependent on these concerns being addressed.

4. We note that Figures 1 and 3 in your submission contain [map/satellite] images which may be copyrighted. All PLOS content is published under the Creative Commons Attribution License (CC BY 4.0), which means that the manuscript, images, and Supporting Information files will be freely available online, and any third party is permitted to access, download, copy, distribute, and use these materials in any way, even commercially, with proper attribution. For these reasons, we cannot publish previously copyrighted maps or satellite images created using proprietary data, such as Google software (Google Maps, Street View, and Earth). For more information, see our copyright guidelines: http://journals.plos.org/plosone/s/licenses-and-copyright.

a)    You may seek permission from the original copyright holder of Figures 1 and 3 to publish the content specifically under the CC BY 4.0 license.  

Reviewers' comments:

Reviewer's Responses to Questions

**Comments to the Author**

1. Is the manuscript technically sound, and do the data support the conclusions?

Reviewer #1: Partly

Reviewer #2: Yes

2. Has the statistical analysis been performed appropriately and rigorously? 

Reviewer #1: No

Reviewer #2: Yes

3. Have the authors made all data underlying the findings in their manuscript fully available?

Reviewer #1: No

Reviewer #2: No

4. Is the manuscript presented in an intelligible fashion and written in standard English?

Reviewer #1: Yes

Reviewer #2: Yes

5. Review Comments to the Author

Reviewer #1: This study attempted to describe the burden and factors associated with MDR-TB in Tigray, Ethiopia. I notified several things that should be clarified by the authors:

1. The study is a cross-sectional study, which is not adequate to analyse factor associated with MDR-TB. Temporality degree should be clearly defined between the factors and outcomes. Since the factors and outcome were identified at the one point in the cross-sectional study, temporality degree is not precise. I suggest changing the term “factors” to “characteristics” of MDR-TB patients.

2. I read that the study defined Tigray, Ethiopia, as the target population and sixth public hospitals were included as the source population in this study. The question is about the adequateness of the sample size. I read that there are many hospitals (private and public) in Tigray and around thousands of people were identified as suspected TB patients. How can you ensure 300 participants and the only public hospital included will represent the TB population in Tigray?

3. Although you explained the sample size calculation, it is still not clear for me. How you defined the sample size?. There is no sampling frame described and if we look at the figure 2, it only 4.2% suspected TB patient who had positive Gen expert. It is a very small proportion. In another side, your source population is the presumptive TB patients. Can you explain that?

4. Can you explain more detail why your data have a high proportion of negative gen expert test? Can you provide detail information about the excluded participants? And how you deal with the representativeness of your data?

5. Please follow standard reporting for observational study (e.g. STROBE)

6. One of your inclusion criteria is “new PTB patient” (line 120), but in the result, you provide data about previously treated TB patients (table 2)?

7. No clear information on how missing data was managed. Please describe it

Abstracts

8. The significant factors should be described in the result of abstract

9. Subject criteria should be described

Introduction

10. Line 92, reference? Adding evidence from another country is suggested. See PMID: 30394362, PMID:31338162.

Method

11. Line 119, please explain the reason to choose the selected hospital.

12. Line 122, the excluded participants should be reported in the result section? How many ? and what is the reason?

13. Line 166, how many TB-HIV patient?

14. Line 236, in which category patient with drug-resistant but not MDR-TB (e.g. mono resistance, IH-R, S-R, any R, etc.)?

Result

15. Line 264, it’s should be tested statistically, and the next is focus on the description only for the included participant

16. Table 1 it’s difficult to understand. Please revise

17. Table 3-5 can be combined. Did you analyse it separately? It should be interpreted in combination.

18. Line 331-343. Please ensure that you compare with the same duration and unit of measurement

Discussion

19. Some findings are different from the global risk factors of MDR-TB (PMID: 30339803), can you explain? It will guide for the specific problem in Tigray as compare with others and give insight for developing specific intervention in Tigray.

Reviewer #2: The prevalence of MDR TB is found to be very high in new and retreatment cases in the Tigray region in Ethiopia. This is an important result from public health standpoint in terms of allocation of resources and need to intervene to limit spread. It would be relevant for authors to discuss in methods section why the six hospitals were selected from the region and if there is a possibility that the population served by the hospitals may have a higher prevalence or risk of MDR TB, such has presence of hotspots, high HIV prevalence, historically poor TB control, etc. It should be discussed in the discussion why the prevalence of MDR TB is higher in the region than in most of the other parts of the countries.

In the methods, authors should describe the standard of practice in these hospitals and the region before this survey. Do all patients get Xpert or culture? Or with this study, are the patients getting systematic DST? If so, is there posssiblity this could influence the prevalence that was found.

6. PLOS authors have the option to publish the peer review history of their article (what does this mean?). If published, this will include your full peer review and any attached files.

Reviewer #1: No

Reviewer #2: Yes: Kunchok Dorjee

---

## [Author Response · Author response to Decision Letter 0]

8 May 2020

Reply: Thank you for your suggestion. We appreciate your concern, but this map was developed by the authors by taking information from a standard map of the study region and developed the study area map. Geographical position system (GPS) readings were taken for the hospitals to indicate where each hospital is located in the region. If you believe this map has a limited contribution to the paper we can agree to exclude it.

Reply: Comment accepted. During data collection, these hospitals were the only hospitals that started GeneXpert as the primary diagnostic technique to all TB suspected patients from the six main zones of the Region. These hospitals had catchment health institutions around them to send sputum samples to these hospitals for diagnosis of TB by the GeneXpert. Out of the 6922 TB suspected patients, only 300 patients were GeneXpert positive. Hence, we have a strong belief that these may represent the Region, even though it is very difficult to be 100% sure. In Tigray Region, only the public hospitals are serving as TB/MDR-TB diagnostic and treatment centers. We have also discussed the representatively of samples in the manuscript.

---

## [Decision Letter · Decision Letter 1]

9 Jun 2020

PONE-D-20-06975R1

Multidrug-resistant Tuberculosis and Its Associated Factors from Tuberculosis Patients in Tigray Region, Ethiopia: a cross sectional study

PLOS ONE

Dear Dr. Welekidan,

Thank you for submitting your manuscript to PLOS ONE. After careful consideration, we feel that it has merit but does not fully meet PLOS ONE’s publication criteria as it currently stands. Therefore, we invite you to submit a revised version of the manuscript that addresses the points raised during the review process.

Please submit your revised manuscript. If you will need more time than this to complete your revisions, please reply to this message or contact the journal office at plosone@plos.org. Please include the following items when submitting your revised manuscript:

We look forward to receiving your revised manuscript.

Kind regards,

Frederick Quinn

Academic Editor

PLOS ONE

Reviewers' comments:

Reviewer's Responses to Questions

**Comments to the Author**

1. If the authors have adequately addressed your comments raised in a previous round of review and you feel that this manuscript is now acceptable for publication, you may indicate that here to bypass the “Comments to the Author” section, enter your conflict of interest statement in the “Confidential to Editor” section, and submit your "Accept" recommendation.

Reviewer #1: All comments have been addressed

Reviewer #2: All comments have been addressed

2. Is the manuscript technically sound, and do the data support the conclusions?

Reviewer #1: Yes

Reviewer #2: Yes

3. Has the statistical analysis been performed appropriately and rigorously? 

Reviewer #1: Yes

Reviewer #2: Yes

4. Have the authors made all data underlying the findings in their manuscript fully available?

Reviewer #1: Yes

Reviewer #2: Yes

5. Is the manuscript presented in an intelligible fashion and written in standard English?

Reviewer #1: Yes

Reviewer #2: Yes

6. Review Comments to the Author

Reviewer #1: I suggest to change the title to "Characteristics of multidrug-resistant tuberculosis patients in Tigray region, Ethiopia".

Reviewer #2: 1) If the lower risk of DR-TB with alcohol consumption is likely a statistical noise, as the authors acknowledged and which likely is the case, then this may not be highlighted in the abstract.

2) In the discussion in lines 461-468, authors attribute the lack of association of sociodemographic factors to possible effect modification by geography. It is hard to conceive that the effect of age, residence, and education status on MDR TB would be different from region to region in Africa or Ethiopia. Looks like authors are using the term effect modification loosely without underlying reason or a basis. Authors can simply say they did not find association between sociodemographic factors and MDR TB prevalence. It is not necessary scientifically and biologically that MDR TB prevalence should be higher or lower in with increasing age; there are communities where actually MDR TB prevalence is higher in younger age group. There are many factors at play.

3) Lines 466-468: "Besides, our finding is different from the global risk factors review report, and this variation among regionally and globally could be due to Effect modification by geographical area". Please pay attention to grammar.

7. PLOS authors have the option to publish the peer review history of their article (what does this mean?). If published, this will include your full peer review and any attached files.

Reviewer #1: No

Reviewer #2: No

---

## [Author Response · Author response to Decision Letter 1]

11 Jun 2020

Comment accepted. The title is changed from “Multidrug-resistant Tuberculosis and Its Associated Factors from Tuberculosis Patients in Tigray Region, Ethiopia: a cross sectional study” to ". Characteristics of Multidrug-resistant Tuberculosis Isolates from Pulmonary Tuberculosis Patients in Tigray Region, Ethiopia: a cross-sectional study". We have added the term “Pulmonary Tuberculosis” because we did not include extra-pulmonary tuberculosis patients.

---

## [Decision Letter · Decision Letter 2]

25 Jun 2020

PONE-D-20-06975R2

Characteristics of multidrug-resistant tuberculosis isolates from pulmonary tuberculosis patients in Tigray Region, Ethiopia: a cross-sectional study

PLOS ONE

Dear Dr. Welekidan,

Thank you for submitting your manuscript to PLOS ONE. After careful consideration, we feel that it has merit but does not fully meet PLOS ONE’s publication criteria as it currently stands. Therefore, we invite you to submit a revised version of the manuscript that addresses the points raised during the review process.

Please submit your revised manuscript. If you will need more time than this to complete your revisions, please reply to this message or contact the journal office at plosone@plos.org. Please include the following items when submitting your revised manuscript:

We look forward to receiving your revised manuscript.

Kind regards,

Frederick Quinn

Academic Editor

PLOS ONE

Reviewers' comments:

Reviewer's Responses to Questions

**Comments to the Author**

1. If the authors have adequately addressed your comments raised in a previous round of review and you feel that this manuscript is now acceptable for publication, you may indicate that here to bypass the “Comments to the Author” section, enter your conflict of interest statement in the “Confidential to Editor” section, and submit your "Accept" recommendation.

Reviewer #1: All comments have been addressed

Reviewer #2: All comments have been addressed

2. Is the manuscript technically sound, and do the data support the conclusions?

Reviewer #1: Yes

Reviewer #2: Yes

3. Has the statistical analysis been performed appropriately and rigorously? 

Reviewer #1: Yes

Reviewer #2: Yes

4. Have the authors made all data underlying the findings in their manuscript fully available?

Reviewer #1: Yes

Reviewer #2: Yes

5. Is the manuscript presented in an intelligible fashion and written in standard English?

Reviewer #1: Yes

Reviewer #2: Yes

6. Review Comments to the Author

Reviewer #1: Regarding the title, I did not find that the authors described and discussed about charactersitics of isolate from MDRTB patient. If so, the manuscript should describe, discuss and explain more detail about microbiological aspects of the isolates. I read that the authors focused on the characteristics of the patients. Therefore, I suggest that the title should be revised to "Characteristics of pulmonary multidrug-resistant tuberculosis patients in Tigray Region, Ethiopia: a cross-sectional study". It will help the readers to understand the manuscript easily.

Reviewer #2: (No Response)

7. PLOS authors have the option to publish the peer review history of their article (what does this mean?). If published, this will include your full peer review and any attached files.

Reviewer #1: No

Reviewer #2: No

---

## [Author Response · Author response to Decision Letter 2]

26 Jun 2020

Comment accepted. The title is changed from “Characteristics of multidrug-resistant tuberculosis isolates from pulmonary tuberculosis patients in Tigray Region, Ethiopia: a cross-sectional study” to "Characteristics of pulmonary multidrug-resistant tuberculosis patients in Tigray Region, Ethiopia: a cross-sectional study".

---

## [Decision Letter · Decision Letter 3]

7 Jul 2020

Characteristics of pulmonary multidrug-resistant tuberculosis patients in Tigray Region, Ethiopia: a cross-sectional study

PONE-D-20-06975R3

Dear Dr. Welekidan,

We’re pleased to inform you that your manuscript has been judged scientifically suitable for publication and will be formally accepted for publication once it meets all outstanding technical requirements.

Kind regards,

Frederick Quinn

Academic Editor

PLOS ONE

Additional Editor Comments (optional):

Reviewers' comments:

Reviewer's Responses to Questions

**Comments to the Author**

1. If the authors have adequately addressed your comments raised in a previous round of review and you feel that this manuscript is now acceptable for publication, you may indicate that here to bypass the “Comments to the Author” section, enter your conflict of interest statement in the “Confidential to Editor” section, and submit your "Accept" recommendation.

Reviewer #1: All comments have been addressed

2. Is the manuscript technically sound, and do the data support the conclusions?

Reviewer #1: Yes

3. Has the statistical analysis been performed appropriately and rigorously? 

Reviewer #1: Yes

4. Have the authors made all data underlying the findings in their manuscript fully available?

Reviewer #1: Yes

5. Is the manuscript presented in an intelligible fashion and written in standard English?

Reviewer #1: Yes

6. Review Comments to the Author

Reviewer #1: (No Response)

7. PLOS authors have the option to publish the peer review history of their article (what does this mean?). If published, this will include your full peer review and any attached files.

Reviewer #1: No